# Clinical and Genetic Management of a Patient with Rubinstein–Taybi Syndrome Type 1: A Case Report

**DOI:** 10.3390/genes16080910

**Published:** 2025-07-29

**Authors:** Victor Santos, Pedro Paulo Chaves de Souza, Talyta Campos, Hiane Winterly, Thaís Vieira, Marc Gigonzac, Alex Honda, Irene Pinto, Raffael Zatarin, Fernando Azevedo, Anna Nascimento, Cláudio da Silva, Aparecido da Cruz

**Affiliations:** 1Graduate Program in Genetics and Molecular Biology, Federal University of Goiás, Goiânia 74605-050, GO, Brazil; victorcortazio@discente.ufg.br (V.S.); pedrosouza@ufg.br (P.P.C.d.S.); campostalytabmd@gmail.com (T.C.); iplazapinto@gmail.com (I.P.); fernandoazevedooncologia@gmail.com (F.A.); 2Replicon Research Nucleus, Graduate Program in Genetics, School of Medical and Life Sciences, Pontifical Catholic University of Goiás, Goiânia 74605-050, GO, Brazil; biomed.hiane.teixeira@gmail.com (H.W.); thaiscidalia@gmail.com (T.V.); marcgigonzac@yahoo.com.br (M.G.); alxhonda@yahoo.com.br (A.H.); dasilva.genetica@gmail.com (C.d.S.); 3Clinical Genetics Service, Center for Rehabilitation and Readaptation Dr. Henrique Santillo, State Health Department of Goiás, Goiânia 74605-050, GO, Brazil; geneticaraffael@gmail.com (R.Z.); karolinne.nascimento@gmail.com (A.N.); 4Undergraduate Program, Academic Institute of Health and Biological Science, State University of Goiás, Goiânia 74690-900, GO, Brazil

**Keywords:** CREBBP, uncommon disease, facial dysmorphisms, broad thumbs, ID, GDD

## Abstract

Rubinstein–Taybi Syndrome type 1 (RSTS1) is an uncommon autosomal dominant genetic disorder associated with neurodevelopmental impairments and multiple congenital anomalies, with an incidence of 1:100,000–125,000 live births. The syndrome, caused by de novo mutations in the CREBBP gene, is characterized by phenotypic variability, including intellectual disability, facial dysmorphisms, and systemic abnormalities. The current case report describes a 15-year-old Brazilian female diagnosed with RSTS1 through whole-exome sequencing, which identified a de novo heterozygous missense mutation in the CREBBP gene (NM_004380.3; c.4393G > C; p.Gly1465Arg), classified as pathogenic. The patient’s clinical presentation included facial dysmorphisms, skeletal abnormalities, neurodevelopmental delay, psychiatric conditions, and other systemic manifestations. A comprehensive genetic counseling process facilitated the differential diagnosis and management strategies, emphasizing the importance of early and precise diagnosis for improving clinical outcomes. This report contributes to the growing knowledge of the genotype–phenotype correlations in RSTS1, aiding in the understanding and management of this uncommon condition.

## 1. Introduction

Rubinstein–Taybi Syndrome (RSTS; OMIM #180849; OMIM #613684) is an uncommon genetic disorder associated with multiple congenital anomalies and neurodevelopmental disorders, with an incidence of 1:100,000–125,000 live births. The main clinical phenotypes of RSTS include short stature, facial dysmorphisms, broad thumbs and halluces, polydactyly, palpebral fissures, low columella, high-arched palate, mask-like face, talon cusps, ptosis, and epicanthus [1]. Individuals with RSTS may present phenotypic variability, particularly regarding the degree of multisystemic involvement, which can affect the respiratory, cardiac, genitourinary, ocular, auditory, orthopedic, endocrine, neurological, dental, and cutaneous systems. Additionally, individuals affected by RSTS exhibit a predisposition to the proliferation of benign tumors [1,2,3,4]. Neurological involvement in RSTS is evident, with highly variable phenotypic features among patients. These include global developmental delay (GDD), intellectual disability (ID), most commonly in the moderate to severe range, with reported IQ values often between 35 and 50, autistic behavior, hypotonia, hyperreflexia, electroencephalogram abnormalities, and agenesis of the corpus callosum. However, cognitive impairment can vary among patients, with some presenting milder or more severe degrees of intellectual disability. Moreover, psychiatric disorders are frequently observed in RSTS patients, including attention deficit hyperactivity disorder (ADHD), impairment in social interaction, and emotional lability [4]. The inheritance pattern of RSTS is autosomal dominant, with 99% of reported cases involving de novo mutations in two genes: CREBBP (RSTS1; OMIM 600140) and EP300 (RSTS2; OMIM 602700). The CREBBP gene is located on cytoband 16p13.3 and encodes the CREB Binding Protein (CBP) (NM_600140). The EP300 gene is located in region 22q13 and encodes the EA1 Binding Protein p300 or p300 (NM_602700). These two genes are similar in function and biological importance. Both are ubiquitously expressed and regulate gene expression through their acetyltransferase activity. Functionally, CBP and p300 transfer acetyl groups (CH_3_CO) to lysine residues on histones and other proteins. Histone acetylation opens chromatin, making it more accessible to the transcription machinery, while the acetylation of transcription factors and regulatory proteins modulates their expression [5,6].

Interestingly, missense mutations involving the terminal portion of exon 30 and the beginning of exon 31 in both the CREBBP and EP300 genes are associated with a distinct phenotype; Menke–Hennekam Syndrome. Individuals with Menke–Hennekam Syndrome do not exhibit the characteristic facial features of RSTS or broad/angled thumbs and halluces. RSTS phenotypes show facial features including ptosis, telecanthus, short and upward-slanting palpebral fissures, depressed nasal bridge, short nose, anteverted nostrils, short columella, and a long philtrum. Other features include short stature, intellectual disability, microcephaly, feeding difficulties, seizures, autistic behavior, and variable additional manifestations [7,8,9]. A search conducted in March 2024 in the DECIPHER database identified 165 reported cases of mutations involving CREBBP and 122 involving EP300, without specifying the phenotypes (RSTS1, RSTS2, or Menke–Hennekam) associated with these mutations [10].

In 2010, RSTS1 represented approximately 50–70% of RSTS cases, while RSTS2 accounted for only 3%, often associated with de novo variants. However, to date, approximately 600 individuals with pathogenic variants in the CREBBP and EP300 genes have been reported [11,12,13,14]. Generally, mutations involving CREBBP, related to the RSTS1 subtype, are not only more frequent but also have a more critical impact on development, resulting in a more severe phenotype compared to the RSTS2 subtype. Exclusive manifestations of RSTS1 include congenital heart disease, genitourinary abnormalities, recurrent respiratory infections, ophthalmological complications, and skeletal malformations [1,10].

Genetic counseling for families of RSTS patients is essential, as it assists with genetic diagnosis, identifies phenotypic variability, assesses and discusses the risk of transmission to offspring, and provides valuable and necessary guidance regarding surveillance and multidisciplinary care required for managing RSTS complications [4,15]. In this context, the present study aims to report a case of RSTS1 addressed in the genetic counseling process, describing its genotypic–phenotypic correlations and a patient’s clinical peculiarities. The description of singular cases is crucial for a better understanding of patient prognosis, inheritance patterns—which enable tailored genetic counseling for each family—treatment, follow-up, and case management, ultimately improving the quality of life for patients and their families.

## 2. Case Report

A female proband, 15 years old, from Goiânia, Central Brazil, was referred to the Genetic Counseling (GC) service at the State Center Rehabilitation and Readaptation (CRER) due to an undiagnosed uncommon disease presenting with multiple neurological abnormalities and some congenital malformations. The proband is the second child of non-consanguineous, healthy parents, aged 30 (mother) and 41 (father) at conception. In the preconception period, the parents experienced reproductive difficulties, leading them to seek assistance at a human reproduction clinic, although initially without success. Later, they managed to conceive naturally. During pregnancy, regular prenatal care was performed. All ultrasounds scheduled in the prenatal period were completed, with no clinically relevant findings. The mother took all the recommended prenatal vitamins. A single episode of maternal bleeding was recorded, which received appropriate medical treatment. There were no reports of infections, amniotic fluid loss, skin rashes, fever, maternal hypertension, or edema. The mother denied exposure to teratogenic agents during pregnancy. Fetal movements were noted around the 18th week; the fetus was not as active as in a previous pregnancy but continued to move until the end of the pregnancy. Maternal weight gain was 15 kg during gestation.

The proband was born at term at 39 weeks of gestation by cesarean delivery due to a medical decision. At birth, her length was 47 cm (p.4%; −1.72 SD), weight was 3100 g (p.25%; −0.67 SD), and head circumference was 35 cm (p.64%; +0.37 SD), with an Apgar score of 9/10. She delayed the newborn reflexive cry. The proband remained in the maternity ward for three days before being discharged, and by the fifth day of life, she presented with myotonic reflexes that persisted for approximately fifteen days. At that time, the parents sought pediatric neurology services. Subsequently, she exhibited significant delays in all developmental milestones, according to pediatric developmental standards. She experienced recurrent infantile ear and throat infections, as well as pneumonia, treated with antibiotics. Over time, the infections became less frequent. Pubertal development began, and menarche occurred at age 10. Currently, based on medical recommendations, menstrual suppression is currently achieved using a contraceptive implant to inhibit the menstrual cycle.

The child has good social interaction, good dietary acceptance, and relative independence for her daily life activities. She has cognitive challenges, particularly in reading, mathematics, and written expression. She demonstrated attachment to certain toys and objects, using them as elements of emotional security and comfort. During early adolescence, at the time of COVID-19 pandemics, she engaged in remote schooling but experienced regression in previously acquired skills, including delays in gross motor development and language acquisition. Additionally, she exhibited impaired social reciprocity, which contributed to the infantilization of her behavior.

During case management and observation of complementary exams requested by the attending multidisciplinary team, several phenotypes were identified in the patient (Figure 1). Facial features included ocular hypertelorism, strabismus, broad nasal base with depressed nasal bridge, high-arched palate, low-set ears, small mouth, prominent eyebrows, and a mask-like face. Skin findings included melanocytic nevi, café-au-lait spots, atypical scarring, and keloid formation. Additionally, neurological abnormalities were identified, including bruxism, neuropsychomotor developmental delay, irritability, anxiety, attention deficit, hyperactivity, impulsivity, sibling-restricted aggressiveness, irrational fears, mood instability, repetitive questioning, agoraphobia, obsessive–compulsive disorder, onychophagia of hands and feet, and soliloquies.

A cranial MRI revealed a millimetric gap at the anterior extremity of the right caudate nucleus, along with a reduced hypotrophic encephalic volume. There was diffuse cortical hypotrophy atypical for her age, widening of the subarachnoid spaces in the middle cranial fossae and frontal convexities, and mild delays in cerebral myelination, including hypomyelination of the anterior limbs of the internal capsules. Moreover, there was a hypotrophic corpus callosum.

The patient’s autistic behavior was evident. In 2022, a behavioral assessment classified the patient as having mild to moderate autism using the Childhood Autism Rating Scale 2 (CARS2, score = 30.5) and an age-compatible diagnosis of ≥ 10 years on the Autism Diagnostic Interview-Revised (ADI-R). Both tools highlighted qualitative impairments in social interaction and communication, along with restricted, repetitive, and stereotyped behavior patterns.

Skeletal abnormalities included pectus carinatum, joint hyperflexibility, short arms, broad fingers, micrognathia, and bilateral clinodactyly of digitus minimus, along with short, broad toes. Radiography revealed humeral hypotrophy and cranial deformity with a hammered copper appearance. During childhood, the patient exhibited short stature for her age and underwent endocrinological treatment to delay the onset of puberty and promote growth, showing slow growth compared to children of the same age. At her last GC consultation, at age 16, the proband’s anthropometric measurements were as follows: weight of 71 kg (p.94%; +1.60 SD), height of 160 cm (p.29%; −0.53 SD), and head circumference of 55 cm (p.40%; −0.2 SD).

The patient receives regular multidisciplinary follow-ups, with weekly psychology and psychopedagogy sessions. This approach has maintained good behavior and peer interaction, although occasional agitation crises still occur. She continues with daily prescriptions of fluoxetine (20 mg) and sertraline (50 mg), which are used to manage anxiety and irritability, symptoms that have been consistently observed during her clinical follow-up.

Based on the family history and phenotypic synopsis, several diagnostic hypotheses were proposed and subsequently evaluated through genetic testing, requested by the medical geneticist and the genetic counselor following parental consent (Table 1).

WES preparation and enrichment were performed using the Illumina DNA Prep with Enrichment kit, and target region capture was carried out with the Twist Biosciences Custom V3 kit, including probes for all exonic and adjacent regions. WES was performed using the NovaSeq 6000 platform. DNA alignment was referenced against version GRCh38 of the human genome using the BWA-MEM program. A de novo heterozygous variant was identified at chr16:3,738,560 (NM_004380.3; c.4393G > C; p.Gly1465Arg) in the CREBBP gene. This resulted in the substitution of the amino acid glycine for arginine. The variant was classified as likely pathogenic (LP) according to the laboratory; however, the same variant has once been reported previously as pathogenic in the ClinVar database. Thus, phenotype and genotype contribute to close the proband’s diagnosis as Rubinstein–Taybi Syndrome 1 (OMIM #600140), ending a 15-year diagnostic odyssey for the family.

According to the family history, the proband represented a simplex case of an autosomal dominant inheritance, with unknown variant penetrance. To date, reported cases of RSTS1 have indicated high penetrance with variable expressivity, even in instances caused by sequence variants [16]. The de novo variant observed in our patient had previously been reported only once in ClinVar as pathogenic. Although the variant observed in the proband and the one reported in ClinVar are located at the same position (c.4393), the nucleotide substitutions differ. The variant previously reported in ClinVar had a G > A substitution, while in the proband a G > C substitution occurred. Despite this difference, both result in the same protein consequence (p.Gly1465Arg) due to the redundancy of the genetic code, thus sharing the same pathogenicity. Additionally, another variant has been reported in ClinVar in a nearby position (c.4394), which also results in the p.Gly1465Arg substitution. Therefore, this case represents the third worldwide report of an alteration involving position 1465 [17]. Thus, we claim that sequence variants at the amino acid residue 1465 in CBP are deleterious to the protein’s functions and are associated with RSTS1. Although variants at this position appear to be highly relevant, they remain rarely reported in publicly available databases. This may be due to a lack of systematic submissions by clinical laboratories and healthcare professionals, or simply due to the rarity of the variant itself. Therefore, we encourage laboratories to report such findings accordingly in order to improve variant classification and expand the collective knowledge of genotype–phenotype correlations in RSTS1.

RSTS1 carriers rarely inherit the condition, as most affected individuals are simplex cases. However, this does not eliminate the possibility of parental germline mosaicism, which carries an empirical risk of <1% for siblings. It is recommended that this information be discussed during the genetic counseling process, particularly in situations where the parents express reproductive interest. For the case reported here, her parents denied reproductive interest for themselves and for their daughter. However, they expressed a desire to know the recurrence risk for their daughter’s offspring in the event of an unexpected future pregnancy. In this context, they were informed of a 50% transmission risk for the variant, corresponding to the Mendelian risk for autosomal dominant inheritance in heterozygosity [4]. Due to the lack of reliable information regarding the penetrance of this variant, the family was informed that, if transmitted, the variant would manifest the RSTS1 phenotype in the carrier.

### Surveillance and Care

Individuals with RSTS1 can present a wide range of associated comorbidities, such as cardiac problems, respiratory disorders, seizures, and a predisposition to infections. Therefore, medical surveillance must be comprehensive and multidisciplinary, with the goal of monitoring and managing the medical as well as progression aspects of the condition. Regular annual follow-ups should include the following specialties: (a) pediatrics, with annual visits for complete physical examinations, growth monitoring, and the evaluation of motor and cognitive development, including the management of autistic behavior; (b) dermatology, with annual consultations to monitor skin conditions; (c) cardiology, with yearly evaluations to detect and manage cardiac issues; (d) ophthalmology, due to the increased risk of glaucoma; (e) otolaryngology, to monitor potential hearing impairments; (f) nephrology, given the risk of renal abnormalities, including cyst development; and (g) oral health should be monitored with dentist consultations every 6 months to address orthodontic problems and manage conditions, such as bruxism [4].

Given the cognitive and behavioral challenges commonly observed in RSTS1, additional support through psychological follow-up, occupational therapy, speech therapy, and music therapy is recommended to promote development and improve communication in addition to adaptive skills. Genetic counseling also plays a key role in confirming the diagnosis, offering recurrence risk information, and supporting family decision-making [1,4].

With proper monitoring and multidisciplinary interventions, individuals with RSTS1 can achieve a life expectancy comparable to that of the general population. Furthermore, to ensure that patients and their families receive the necessary support, dedicated support groups can provide therapeutic guidance, specialized educational services, and family assistance. These initiatives significantly improve quality of life and reduce the sense of isolation, often experienced due to the low prevalence of RSTS1 in the population [1,4].

## 3. Discussion

The proband exhibited multisystem impairments typical of RSTS1, which can be explained by the multifaceted functions of the CBP protein. CBP acts as a transcriptional co-activator for various transcription factors, regulates gene expression through chromatin remodeling, and is essential for proper embryonic development by controlling cellular processes such as growth, differentiation, and homeostasis. CBP haploinsufficiency can result in significant phenotypic variability among carriers, as observed in our proband [4,18]. She presented four clinical phenotypes not included in the HPO syndromic spectrum for RSTS1, namely melanocytic nevi, pectus carinatum, bruxism, and encephalic hypotrophy.

Regarding the spectrum of mutations in the CREBBP gene associated with RSTS1, among the 165 reported cases of the syndrome in the DECIPHER database, 68 out of 165 (41.2%) corresponded to single-nucleotide variants (SNVs), and 95 out of 165 (58.8%) were copy number variants (CNVs). Additionally, SNVs and CNVs are of de novo origin in 63% and 36% of cases, respectively. Large CNVs are often inherited (27%) from parents carrying balanced rearrangements [10]. The present case is characterized by a de novo SNV, which corresponds to the smaller portion of the mutation profile in individuals with RSTS1.

Genetic evaluation for individuals with suspected RSTS1 is essential since the phenotypes associated with this syndrome are not exclusive and may overlap with other syndromes. This phenotypic overlap often occurs among Mendelian disorders affecting genes involved in chromatin structure and remodeling, known as chromatinopathies. These disorders include Floating–Harbor syndrome (FHS) (OMIM #136140), Cornelia de Lange syndrome (CDLS) (OMIM #122470, #300590, #610759, #614701, #300882, #608749), Wiedemann–Steiner syndrome (WDSTS) (OMIM #605130), Kabuki syndrome (OMIM #147920, #300867), Genitopatellar syndrome (OMIM #606170), Biesecker–Young–Simpson syndrome (OMIM #603736), and Gabriele-de Vries syndrome (OMIM #617557) [18].

A literature review analyzed the main clinical phenotypes in 308 reported cases of RSTS1 [18]. Among the 30 highlighted phenotypes, the proband exhibited 25 of them and did not exhibit 5, namely microcephaly, hypertrichosis, keloids, cardiac anomalies, constipation, and urinary tract anomalies (Table 2). The 83% overlap with the main phenotypes of RSTS1 reinforces the proband as being a typical case of RSTS1. On the other hand, some phenotypes presented by the proband were not identified as the most common ones among the 308 reviewed cases, specifically pectus carinatum, café-au-lait spots, bruxism, feeding difficulties, gastroesophageal reflux, respiratory tract infections, prominent forehead, joint hypermobility, hypotonia, and ptosis.

Genetic counseling for individuals with suspected RSTS1 is critical as it allows for detailed phenotyping, addressing the syndrome’s phenotypic variability. Furthermore, it discusses and provides guidance on the most appropriate genetic tests, which are essential for a differential genetic diagnosis, and identifies at-risk family members. The process also includes genetic education on the inheritance pattern as well as the medical, psychological, and familial implications of the disease, providing key information for medical surveillance and therapies that can enhance the quality of life of affected individuals [15]. The family assisted at our institution received a comprehensive report, which systematically documented and consolidated all pieces of information discussed during the genetic counseling to ensure that current and future professionals assisting the proband could have a clear understanding of her genetic condition and its clinical implications.

## 4. Conclusions

This case report underscores the importance of early and accurate diagnosis in patients with complex and multisystemic phenotypic features, such as those observed in Rubinstein–Taybi Syndrome Type 1 (RSTS1). The use of advanced genomic tools, such as whole-exome sequencing (WES), was essential for identifying the de novo pathogenic variant in the CREBBP gene, thereby confirming the diagnosis after a prolonged diagnostic odyssey and guiding clinical management. Genetic counseling proved to be an indispensable component of the patient’s care, not only facilitating a clearer understanding of the inheritance pattern and reproductive risks but also offering essential guidance for managing the patient’s clinical needs. With a multidisciplinary approach to treatment and surveillance, including regular follow-ups across various specialties, it is possible to optimize care, address comorbidities, and improve the overall quality of life for individuals with RSTS1. This case also underscores the value of detailed phenotypic assessment and genetic testing in guiding clinical decisions and enhancing patient management in uncommon genetic disorders.

## Figures and Tables

**Figure 1 genes-16-00910-f001:**
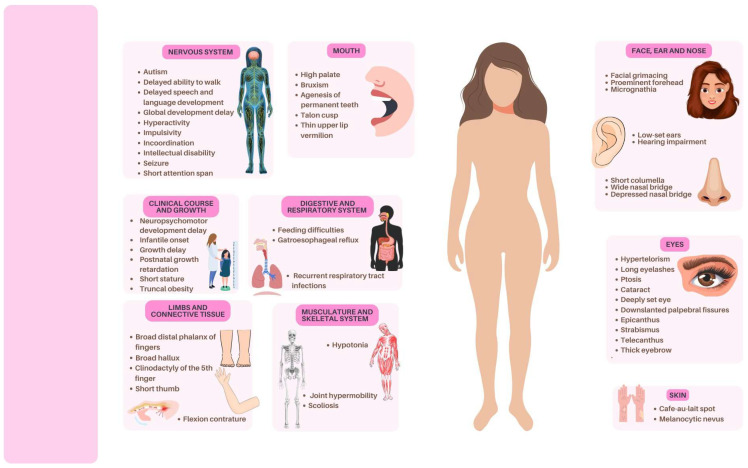
Phenotypic synopsis according to the human phenotype ontology found in a patient with Rubinstein–Taybi Syndrome Type 1 (OMIM #600140) harboring a disease-causing variant (c.4393G > C) in the CREBBP gene.

**Table 1 genes-16-00910-t001:** Follow-up genetic testing for a patient with early myotonic reflexes, global developmental delay, and autism behavior, and a hypotrophic corpus callosum.

Methodology	Result	Interpretation
GTG-Karyotyping	46, XX	No numerical or structural abnormalities found
mFISH for Pallister–Killian Syndrome	Negative	No isochromosome 12p detected
Genotyping of *FMR1* by capillary PCR	Alleles with 25 and 28 CGG repeats	Expansion within normal range
Chromosomal microarray analysis	No genomic gains or losses	Absence of clinically relevant CNVs
Sequencing of *ZNF148* and *KALRN*	No sequence variants found	No clinically relevant genomic variants
Whole-exome sequencing (WES)	Likely pathogenic heterozygous variant: NM_004380.3 c.4393G > C p.Gly1465Arg	Rubinstein–Taybi Syndrome 1 (OMIM #600140)

**Table 2 genes-16-00910-t002:** Phenotypic assessment and comparison between a 15-year-old girl diagnosed with Rubinstein–Taybi Syndrome 1 (OMIM #600140) in Central Brazil and a cohort of 308 patients [18].

Phenotypes	HPO ID	Lacombe et al. (2024) [18]	Proband
**Growth**			
Intrauterine growth retardation	1511	49	Yes
Postnatal growth retardation	4322	75	Yes
Obesity	1513	29	Yes
Microcephaly	252	54	No
**Craniofacial Features**			
Highly arched eyebrows	2253	85	Yes
Long eyelashes	527	89	Yes
Epicanthal folds	286	44	Yes
Strabismus	486	71	Yes
Myopia	545	56	Yes
Downslanted palpebral fissures	494	79	Yes
Convex nasal ridge	444	81	No
Columella below alae nasi	9765	88	Yes
Typical smile	273	94	Yes
Highly arched palate	2705	77	Yes
Talon cusps	11087	73	Yes
Micrognathia	347	61	Yes
Low-set ears	369	44	Yes
**Trunk and Limbs**			
Broad thumbs	11304	96	Yes
Angulated thumbs		49	Yes
Broad fingertips	11300	87	Yes
Broad halluces	10055	95	Yes
Hypertrichosis	998	76	No
Keloids	10562	23	No
Scoliosis	2650	18	Yes
Cardiovascular anomalies	2564	35	No
Constipation	2019	76	No
Urinary tract anomalies	79	28	No
**Neuromuscular**			
Seizures	1250	25	Yes

## Data Availability

All data relevant to the current study were included in this manuscript and in DECIPHER.

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
