# Peer review of "Clinical and Genetic Management of a Patient with Rubinstein–Taybi Syndrome Type 1: A Case Report"

_genes, 2025, doi:10.3390/genes16080910_

Round 1
Reviewer 1 Report
Comments and Suggestions for Authors
Interesting work, limited by its description of a single case. The pathology is rare, but the paper could be enriched with a literature review, reporting the described cases with phenotype-genotype correlations. The scientific English language needs to be revised.
Author Response
Comments 1: L. 30 In the Graphical Abstract, authors write “A human neurodevelopmental disorder” under the Results column. The use of “human” is unnecessary when referring to neurodevelopmental disorders. Response 1: I would like to thank you for the valuable comments, which have been fully addressed. I have removed the word “human” from the term “human neurodevelopmental disorder” in the Graphical Abstract. Additionally, as suggested in Comment 3, I have replaced “ultra-rare disease” with “uncommon disease.”
|
Comments 2: L. 43-44 “with an intelligence quotient (IQ) ranging from 35-50” – there is variability to the IQ of these children and that should be addressed |
Response 2: We agree. We rewrote the paragraph: “These include global developmental delay (GDD), intellectual disability (ID), most commonly in the moderate to severe range, with reported IQ values often between 35 and 50, autistic behavior, hypotonia, hyperreflexia, electroencephalogram abnormalities, and agenesis of the corpus callosum. However, cognitive impairment can vary among patients, with some presenting milder or more severe degrees of intellectual disability.”
Comments 3: This paper describes the condition as “rare” and “ultrarare” throughout. In the U.S., this is not considered a rare disorder. Replace this with more appropriate terminology, such as “uncommon”.
Response 3: According to the OMIM and GeneReviews databases, the estimated prevalence of Rubinstein-Taybi syndrome type 1 (RSTS1) is approximately 1 in 100,000 to 1 in 125,000 live births. In Brazil, this condition is classified as a rare disease based on national criteria.
We would appreciate clarification on whether RSTS1 is officially considered an uncommon disease rather than a rare disease in the United States. If so, we are happy to revise and standardize the terminology throughout the manuscript to reflect this classification and align with the reviewer’s recommendation.
Comment 4: L. 47 “good social interaction” does not fit in the listed psychiatric disorders frequently observed in RSTS patients. This paragraph needs to be reworked addressing both issues.
Response 4: We agree. It was written incorrectly. The correct term should be “impairment in social interaction.”
Comment 5: L. 43-44 Please provide a references to confirm IQ range of RSTS patients as 35-50 and address the range of IQs in the disorder
Response 5: The reference is from the OMIM/GeneReviews database (reference number 4). The patient’s IQ is not available in the medical records. So far, we only have a clinical diagnosis of intellectual disability, established by the neurologist. I am currently reaching out to the patient’s mother to obtain further information. If successful, I will include it prior to the completion of the submission process.
Comment 6: L. 117-124 This paragraph suggests that the patient is functioning at a higher level than someone with an intellectual disability and instead presents more characteristics of a language learning disorder. This is why the variability of the IQ and is she first reported case or one of many cases.
Response 6: Thank you for your thoughtful observation. The patient presents with mild intellectual disability, which is indeed outside the expected phenotypic spectrum for Rubinstein-Taybi syndrome type 1 (RSTS1), where most reported cases range from moderate to severe intellectual disability.
In the medical records, the patient initially received the ICD-10 diagnosis of unspecified intellectual disability (F79). However, after a comprehensive evaluation by a medical geneticist and genetic counselors, we concluded that the clinical presentation is more consistent with mild intellectual disability.
This milder cognitive profile may represent an example of phenotypic variability within RSTS1.
We believe that the observed learning disorder may, in fact, reflect an underlying mild intellectual disability. Given the clinical evaluation and developmental history, it is likely that the learning difficulties are part of the broader cognitive profile associated with the patient’s condition.
Comment 7: L. 162 Please provide more insight on why fluoxetine and sertraline were prescribed, as these are primarily used to treat depression and anxiety, not ADHD.
Response 7: We agree that further clarification is needed in this paragraph regarding the use of fluoxetine and sertraline. As mentioned in lines 132 and 134, the patient presents with anxiety and irritability. Both medications are indicated for the treatment of anxiety disorders. Additionally, fluoxetine, although used off-label, may be prescribed to manage irritability and compulsive behaviors, particularly in individuals with autism spectrum disorder (ASD).
Comment 8: L. 195-195 Please clarify this sentence, “Therefore, laboratories should report it accordingly.”
Response 8: We agree that this sentence required further clarification. A more appropriate and explanatory version would be:
“Although variants at this position appear to be highly relevant, they remain rarely reported in publicly available databases. This may be due to a lack of systematic submission by clinical laboratories and healthcare professionals, or simply due to the rarity of the variant itself. Therefore, we encourage laboratories to report such findings accordingly in order to improve variant classification and expand the collective knowledge of genotype–phenotype correlations in RSTS1.” Additional information will be included in this paragraph of the manuscript to provide proper support for the use of these medications in the clinical management of this patient.
Comment 9: L. 222-228 This paragraph is redundant following the paragraph written above. Authors should consider which paragraph best fits the article and remove the other.
Response 9: Thank you for your observation. We agree with your comment regarding the redundancy between the two paragraphs. To address this, we have revised the second paragraph to remove the overlapping content and focus on complementary aspects of patient management.
A more appropriate and explanatory version would be:
Given the cognitive and behavioral challenges commonly observed in RSTS1, additional support through psychological follow-up, occupational therapy, speech therapy, and music therapy is recommended to promote development and improve communication and adaptive skills. Genetic counseling also plays a key role in confirming the diagnosis, offering recurrence risk information, and supporting family decision-making.
We kindly ask if the current version is acceptable to you. |
4. Response to Comments on the Quality of English Language |
Point 1: Dear May Gao suggested via email that the English in the document requires improvement. I would kindly like to request an extension of 15 days in order to allow Professor Aparecido Divino da Cruz to return from his vacation and assist with the English revision of the manuscript. His contribution will be essential to ensure the quality and clarity of the final version. I truly appreciate your understanding and consideration. |
|
5. Additional clarifications |
· The error regarding the affiliations of the author Victor Cortázio do Prado Santos was previously discussed via email. The correction has already been made in the manuscript. · An error was identified in the numbering of references between 4 and 15, and they have been corrected both in the main text and in the reference list." |

Reviewer 2 Report
Comments and Suggestions for Authors
Please see comments in file attached.

Author Response

(The authors gave the same response as above.)
